# The Correlation between Extracellular Heat Shock Protein 70 and Lipid Metabolism in a Ruminant Model

**DOI:** 10.3390/metabo12010019

**Published:** 2021-12-27

**Authors:** Miloš Ž. Petrović, Marko Cincović, Jože Starič, Radojica Djoković, Branislava Belić, Miodrag Radinović, Mira Majkić, Zoran Ž. Ilić

**Affiliations:** 1Faculty of Agronomy, University of Kragujevac, 32000 Čačak, Serbia; petrovic.milos87@kg.ac.rs (M.Ž.P.); radojicadjokovic@gmail.com (R.D.); 2Department of Veterinary Medicine, Faculty of Agriculture, University of Novi Sad, 21000 Novi Sad, Serbia; miodrag.radinovic@polj.uns.ac.rs (M.R.); miramajkic@gmail.com (M.M.); 3Veterinary Faculty, University of Ljubljana, 1000 Ljubljana, Slovenia; joze.staric@vf.uni-lj.si; 4Academy of Medical Science SMA, 11000 Belgrade, Serbia; drbbelic@gmail.com; 5Faculty of Agriculture, University of Priština-Kosovska Mitrovica, 38219 Lešak, Serbia; zoran.ilic@pr.ac.rs

**Keywords:** cow, early lactation, chaperons, lipolysis, ketogensis, insulin resistance, fatty liver, negative energy balance

## Abstract

Metabolic stress in early lactation cows is characterized by lipolysis, ketogenesis, insulin resistance and inflammation because of negative energy balance and increased use of lipids for energy needs. In this study the relationship between lipid metabolite, lipid-based insulin resistance, and hepatocyte functionality indexes and tumor necrosis factor alpha (TNF-α) with extracellular heat shock protein 70 (eHsp70) was investigated. The experiment included 50 cows and all parameters were measured in blood serum. In cows with a more pronounced negative energy balance, the following was determined: a higher concentration of eHsp70, TNF-α, non-esterified fatty acid (NEFA), beta-hydroxybutyrate (BHB), NEFA to insulin and NEFA to cholesterol ratio and lower concentration of cholesterol, very low-density lipoproteins (VLDL), low density lipoproteins (LDL) and liver functionality index (LFI). The eHsp70 correlated negatively with the values of cholesterol, VLDL, LDL, and triglycerides, while correlated positively with the level of NEFA and BHB. A higher concentration of eHsp70 suggests the development of fatty liver (due to a higher NEFA to cholesterol ratio and lower LFI) and insulin resistance (due to a lower revised quantitative insulin sensitivity check index RQUICKI-BHB and higher NEFA to insulin ratio). The eHsp70 correlated positively with TNF-α. Both TNF-α and eHsp70 correlated similarly to lipid metabolites. In cows with high eHsp70 and TNF-α values we found higher concentrations of NEFA, BHB, NEFA to insulin and NEFA to cholesterol ratio and a lower concentration of triglycerides and VLDL cholesterol compared to cows that had only high TNF-α values. Based on the positive correlation between eHsp70 and TNF-α, their similar relations, and the additional effect of eHsp70 (high TNF-α + eHsp70 values) on lipid metabolites we conclude that eHsp70 has pro-inflammatory effects implicating lipolysis, fatty liver, and fat tissue insulin resistance.

## 1. Introduction

Heat shock proteins (Hsp) are molecular chaperones that play a key role in maintaining protein homeostasis in the cell (protostasis) [1]. They prevent protein misfolding and aggregation, which is achieved through their action on folding intermediates [2,3,4]. These proteins are classified based on their molecular mass, and the most important is heat shock protein 70 (Hsp70) with a molecular mass of about 70 kDa, which is designated as the “master player in protein homeostasis” [5]. The Hsp concentration increases significantly when exposed to a stressor originating from the cell itself or from the external environment. Many chaperones are induced under the influence of high ambient temperatures, when the universal heat shock response (HSR) develops, hence the name heat shock proteins [6,7]. Intracellular Hsp70 (iHsp70) shows its protective and anti-inflammatory effects. Induced iHsp70 protects the cell from apoptosis by reducing or blocking caspase activation, binding to apoptosis-inducing factor (AIF), and inhibiting AIF-induced chromatin condensation, preventing mitochondrial damage and nuclear fragmentation [8,9]. It blocks morphological changes in cells caused by tumor necrosis factor-induced apoptosis and has been shown to support cell repair after inflammatory damage [10,11]. The anti-inflammatory effect of iHsp70 is reflected in the fact that it inhibits the response to lipopolysaccharides and blocks the production of inflammatory mediators such as TNF-α, and other mechanisms have been described [12]. Gene expression for Hsp70 production is well studied in ruminants or their cell cultures exposed to high ambient temperatures, and multiple increases of iHsp70 in cells provide better adaptation to heat stress [13,14,15].

Hsp70 has a dual role in the body depending on whether it is located intra- or extracellular, so that iHsp70 has a protective role and eHsp70 has a proinflammatory role. The study of eHsp70 has become relevant due to the availability of diagnostic kits for determining its concentration, and recent results show that it is a very useful predictor of mortality in patients with septic shock [16]. Hsp70 travels to the extracellular space in several ways: through leaking from necrotic cells, under the action of various stressors and inflammation in intact cells, it can be produced in the liver as an acute phase protein, and exosome transport and direct contact with the cell lipid membrane have been described [17,18]. The pro-inflammatory effect of eHsp70 is achieved through the introduction of innate immune cells. It induces the secretion of inflammatory cytokines (TNF-α, IL-1β, IL-6), the expression of inducible nitric oxide synthase (iNOS) and the nuclear translocation of Nuclear Factor-κB (NF-κB) [19]. According to the chaperone balance theory, the higher the value of eHsp70 compared to iHsp70, the more pronounced the pro-inflammatory effects [20].

The peripartal period in dairy cows is a model for the study of metabolic stress. Metabolic stress occurs in cows in early lactation as a result of calving and the onset of lactation, which is steeply rising. Metabolic stress is characterized by a negative energy balance, increased milk production, that directs glucose to the mammary gland while meeting the energy needs of other tissues through lipolysis and ketogenesis that affect the overall metabolic adaptation of cows [21,22]. Increased influx of fatty acids into the liver, with a concomitant decrease in lipoprotein transport in the body, leads to the development of fatty liver and ketosis [23]. The only lipogenic hormone in the body of cows is insulin, the level of which decreases, and its action is inhibited under the influence of pronounced lipolysis and ketogenesis, resulting in insulin resistance, which further stimulates lipolysis [24]. The development of lipolysis, fatty liver and insulin resistance in dairy cows is closely associated with the development of inflammation in early lactation. Indeed, increased lipolysis in cows can lead to the release of larger amounts of pro-inflammatory cytokines from adipose tissue known as adipokines. The most important of these is TNF-α, which has been associated with the development of insulin resistance and increased lipase activity [25]. Increased serum TNF-α concentration has been found in cows with moderate to severe fatty liver syndrome [26]. Several experimental studies have shown a direct influence of NEFA on inflammatory processes by affecting the activation of toll-like receptors (TLRs), particularly TLR4. The activation of TLR4 can lead to an inflammatory response with the release of pro-inflammatory cytokines [27,28]. The period of lactation and milk production in cows affect eHsp70 levels. Thus, it was found that after parturition in early lactation and with increasing milk production, the concentration of eHsp70 in blood serum and saliva increases in cows [29,30,31]. The dynamic changes of eHsp70 and TNF-α concentrations are almost identical in cows during the first weeks after calving when under metabolic stress so that the concentrations of both parameters increase [29]. In addition, Hsp70 affects many metabolic processes related to energy metabolism, liver lipid metabolism, and insulin resistance, and exerts its effects through TLR4 [32,33].

All these suggest that there is an overlap of pathways and regulatory mechanisms between lipid metabolites, eHsp70 and TNF-α, and previous studies, have not investigated their relationship in the blood of cows in early lactation. In this study, the relationship between lipid metabolite, lipid-based insulin resistance, and hepatocyte functionality indexes and tumor necrosis factor-alpha (TNF-α) with extracellular heat shock protein 70 (eHsp70) was investigated, to determine its pro- or anti-inflammatory properties in the first weeks after calving.

## 2. Results

The concentration of eHsp70 in blood serum was 3.25 ± 1.43 ng/mL (1.3–6.0 ng/mL). The concentration of lipid parameters and the values of functionality indices are shown in Table 1 as mean ± standard deviation. In cows with a more pronounced negative energy balance, the following was determined: a higher concentration of eHsp70, TNF-α, non-esterified fatty acid (NEFA), beta-hydroxybutyrate (BHB), NEFA to insulin and NEFA to cholesterol ratio and lower concentration of cholesterol, very low-density lipoproteins (VLDL), low-density lipoproteins (LDL), and liver functionality index (LFI). The eHsp70 correlates negatively with the values of total cholesterol, very low-density lipoproteins (VLDL), low density lipoproteins (LDL), and triglycerides, while it correlates positively with the values of non-esterified fatty acid (NEFA) and beta-hydroxybutyrate (BHB). The eHsp70 positively correlates with NEFA to cholesterol ratio, NEFA to insulin ratio, and negatively with LFI and RQUICKI-BHB. The eHsp70 correlates positively with TNF-α. The correlation coefficient and statistical significance are shown in Table 1.

The regression lines are shown in Figure 1a–l. The linear regression parameter b describes the change in blood parameters for each 1 ng/mL increase in eHsp70. For every 1 ng/mL increase in eHsp70 the concentration of cholesterol decreased by 0.46 mmol/L, triglycerides by 0.03 mmol/L, LDL by 0.43 mmol/L, VLDL by 0.0056 mmol/L, LFI by 2.5 units, and that of RQUICKI-BHB by 0.03 units. In contrast, to these parameters, for every 1 ng/mL eHsp70 increase NEFA concentration increased by 0.09 mmol/L, BHB by 0.06 mmol/L, HDL by 0.047 mmol/L, NEFA to cholesterol ratio by 0.05 units, NEFA to insulin ratio by 0.03 units, and TNF-α concentration by 0.96 ng/mL.

Both TNF-α and eHsp70 correlate identically with lipid metabolites, so the value of the correlation coefficients are similar and the correlation direction is identical (Table 2). The correlation between eHsp70 and triglycerides, NEFA, VLDL, NEFA to cholesterol ratio, NEFA to insulin ratio, and LFI was still significant after exclusion of TNF-α as a control factor, indicating an independent effect of eHsp70 on these parameters. eHsp70 showed a non-independent effect on cholesterol, LDL, BHB, and RQUICKI-BHB because the statistically significant correlation was lost after the exclusion of TNF-α.

In cows with high values of eHsp70 and TNF-α, we found higher concentrations of NEFA, BHB, NEFA to insulin and NEFA to cholesterol ratio and a lower concentration of triglycerides and VLDL cholesterol compared to cows that have only high values of TNF-α (Table 3).

## 3. Discussion

The average concentration of eHsp70 obtained in this experiment is consistent with the results of previous studies. Kristensen et al. [31] found the concentration of this chaperone at 4.46 ng/mL (range 0.24–26.47 ng/mL). Catalani et al. [29] found values in the range of 2.5–4.5 ng/mL while these values were above 7 ng/mL in the later lactation phase. Concentration of eHsp70 in saliva in cows ranged from 0.524 to 12.174 ng/mL [30]. Hsp70 is significantly regulated by high and low ambient temperatures and is a very sensitive marker of heat stress in cows [34]. The cows in this experiment calved in March and April, which are markedly thermoneutral months in our geographic region [35], so the external ambient temperatures did not affect the value of eHsp70. 

In cows with a pronounced negative energy balance, the concentration of eHsp70 increases, and other parameters indicate increased lipolysis, ketogenesis, fatty infiltration of hepatocytes and insulin resistance. An increase in eHsp70 in cows with a pronounced negative energy balance implies that metabolic stress in early lactation leads to increased secretion of this chaperone into the bloodstream, so it can have negative effects. It has been described that Hsp60 increases in the cytosol under the action of various stressors, that they interact with the lipid membrane during prolonged stressors and travel into the bloodstream where they initiate inflammatory processes and atherosclerosis [36]. In autoimmune diseases, increased production of Hsp70 has been shown in the cell affected by the autoimmune process, which causes the immunosuppressive effect of iHsp70. Prolonged action of inflammatory stress in the autoimmune process leads to additional production of Hsp70 which passes into the extracellular part, thus eHsp70 stimulates inflammation and further development of autoimmune disease, with the loss of iHsp70 [37]. 

The interaction of chaperones with lipids has been shown in a number of studies. Hsp70 was found to interact with the lipid membrane showing its affinity for lipids [38]. Also, as eHsp70 increases, LDL decreases in people with acute myocardial infarction [39]. Certain correlations have been established between lipoprotein fractions and Hsp60 [40], while Hsp90 regulates lipid biosynthesis [41]. Although certain connections have undoubtedly been established, in cows it is necessary to analyze the action of chaperones in the function of fatty liver, lipolysis, and insulin resistance as the most important homeoretic process in early lactation.

In early lactation, the most dramatic changes are associated with lipid metabolism. The values of NEFA, BHB, cholesterol, triglycerides, HDL, LDL, and VLDL are in agreement with previously published results [42,43]. The values of LFI and NEFA to cholesterol ratio are comparable to those obtained previously and they are sensitive indicators of the functional status of hepatocytes whose value is based on lipids [44,45]. Lipid metabolism in ruminants is specific and there is a close relationship between the metabolism of fatty acids, triglycerides, cholesterol, and lipoproteins that cross in the liver as the most important metabolic organ [23,46,47,48,49]. NEFAs are a very important component of blood plasma in the process of lipomobilization in the peripartum period in cows. The fatty acids used for the synthesis of liver triglycerides in small part from the endogenous synthesis in the liver from acetyl-Co-A, and in large part from free fatty acids taken from the blood and formed mainly by lipolysis from adipose tissue. The reason for the great variability of their concentration in the blood is their very short half-life of only 2 to 3 min [50,51]. Therefore, free fatty acids are considered to be the most metabolically active component of lipids in the body. The concentration of free fatty acids in the blood is almost 2 times higher in lactating than in dry cows, while the concentration of triglycerides in the blood is completely reversed. It is thought that the increased lipolysis is triggered not only by energy deficits during this period but also by the changes in hormonal status before calving. At the heart of this process is the hydrolysis of triglycerides in adipose tissue. This complex process occurs under the control of lipolytic hormones that activate c-AMP, which initiate hormone-sensitive lipoprotein lipase. Based on measurements of palmitic acid flux in venous blood of cows immediately after parturition, it was determined that up to 2.9 kg of fat is mobilized daily from body reserve. Fatty acids are involved in the following metabolic processes: (1) complete oxidation of fatty acids to CO_2_ and H_2_O; (2) partial oxidation of fatty acids to acetyl-Co-A and synthesis of ketone bodies; (3) resynthesis (reesterification) of fatty acids to triglycerides and their transport from the liver with VLDL fraction of lipoproteins; (4) resynthesis of triglycerides and their retention in the liver with the possibility of fatty infiltration of hepatocytes. The liver is capable of extracting about 30 percent of free fatty acids from the blood during both starvation and normal nutrition. Most of the fatty acids retained in the liver are decomposed to acetyl-Co-A by the process of β-oxidation or esterified to triglycerides and phospholipids. Acetyl-Co-A is further oxidized in the liver to CO_2_ and H_2_O or ketone bodies. Triglycerides can be transported with the blood via the hepatic vein in the form of VLDL and other lipoprotein fractions. For lipid metabolism to proceed smoothly, an appropriate “pool” of transport lipoprotein systems is required to enable the transport of hydrophobic lipid molecules through the aqueous medium of the blood plasma. Lipoproteins are synthesized from apoproteins and lipids in the cisterns of the granulated reticulum and the Golgi zone of hepatocytes. Due to the low presence of chylomicrons in ruminants blood plasma, the main transport system for triglycerides from enterocytes to the liver, and from the liver to extrahepatic tissues, is the VLDL fraction of lipoproteins. The half-life of chylomicrons and VLDL fractions of lipoproteins in the blood plasma of ruminants is very short, ranging from only 2 to 11 min, which is one of the reasons for their very low concentration in the blood. Therefore, HDL fractions accounted for 70 percent and LDL fractions for 20 percent of total blood lipoprotein [52,53]. Triglycerides make up the highest percentage in the VLDL fraction, so the role of the liver in the synthesis of this fraction is of great importance in triglyceride metabolism. Total lipids normally account for about 5 percent of the total liver mass in cows. Although the synthesis of triglycerides in the liver is very active in the process of esterification of free fatty acids originating from adipose tissue, the amount of triglycerides in the liver is small, because VLDL can transport newly synthesized triglycerides from the liver in physiological conditions. It is believed that triglycerides do not normally accumulate in the liver because they are transported from the liver at the same rate at which they are synthesized. Therefore, the role of the liver in the synthesis of lipoproteins is crucial for the normal functioning of lipid metabolism in the body. According to some data, cows have a limited ability of liver cells to synthesize the VLDL fraction of lipoproteins, putting them at high risk of accumulating triglycerides in the liver leading to fatty infiltration and degeneration of hepatocytes. It was also found that cattle have markedly reduced ability to secrete the VLDL fraction of liver lipoprotein during starvation. In cows with hepatic lipidosis, the release of triglycerides from the liver is reduced, and therefore, the concentration of triglycerides rich lipoproteins in the blood serum is very low. In contrast to humans, in which cholesterol synthesis in the intestines is very important, in animals, the largest amounts of cholesterol are synthesized in the liver. After synthesis in hepatocytes, cholesterol is released into the blood in the form of lipoproteins. Most cholesterol is found in the LDL fraction (β-lipoproteins), which is formed from the VLDL fraction. Since the basic synthesis of cholesterol occurs in the liver, in cows with impaired liver function there is a disruption in synthesis. Indeed, several authors have determined a decrease in blood cholesterol concentration in the peripartum period in high-yielding cows that have a pronounced energy deficit. In high-yielding cows, there is a significant decrease in blood cholesterol concentration in the peripartum period compared with values obtained 2 to 3 months before parturition and 1 to 2 months after parturition. These changes are particularly pronounced in animals in which fatty infiltration and degeneration of hepatocytes were established. 

Based on the peculiarities of lipid metabolism in cows, we conclude that an increase in eHsp70 and TNF-α increases lipolysis and ketogenesis and decreases the production of LDL and VLDL with a decrease in triglycerides and cholesterol and declining liver function through declining LFI and NEFA to cholesterol ratio. All these correlations suggest that eHsp70 is associated with the formation of a fatty liver in early lactating cows. An in vitro study has shown that Hsp70 has a beneficial effect on liver lipogenesis in subjects with non-alcoholic fatty liver disease (NAFLD) fed a high-fat diet [54]. Induction of Hsp70 in the liver improves lipid metabolism, while loss of Hsp70 leads to a tendency to oxidative liver injury and lipidosis [32]. Because early lactation and ketotic cows have increased apoptosis of hepatocytes [55,56], it is possible that there is a loss of Hsp70, which turns into eHsp70, which reduces its protective effects and increases the negative pro-inflammatory effects. There are insufficient data in the current literature on the effect of extracellular Hsp on lipid metabolism in the liver. Elevated TNF-α levels have been found in cows with fatty liver, and administration of recombinant bovine TNF-α (rbTNFα) for 7 days leads to triglyceride accumulation and hepatocyte inflammation with decreased gluconeogenesis [57,58]. 

Lipolysis is the main driver of all metabolic changes in early lactation, and a positive correlation between eHsp70 and TNF-α with NEFA was found as an indicator of lipolysis in cows. Thermal stress is known to translocate Hsp70 into lipid droplets, and proteomic analysis of lipid droplets has shown that they always contain the Hsp70 family of chaperones [59,60]. Heat shock cognate 70 (HSC70) protein is involved in the chaperone-mediated autophagy process and regulates lipolysis via perlipin [61,62]. These findings suggest a direct link between Hsp70 and lipolysis, supporting our findings that eHsp70 correlates independently with NEFA values and other derived parameters. On the other hand, adipose tissue remodeling in early lactation cows is known to involve the activation of macrophages in adipose tissue, which triggers an inflammatory response by increased secretion of pro-inflammatory cytokines, especially TNF-α [63]. Changes in perilipn expression leads to an inflammatory response of adipose tissue when the TNF-α level increase and these changes are very pronounced in ketotic cows [64]. This could be the reason why TNF-α statistically controls the correlation between eHsp70 and BHB and RQUICKI-BHB. Pronounced lipolysis is directly related to insulin resistance in cows and can significantly affect other aspects of metabolism in early lactation [65,66]. Insufficient production of insulin is associated with its reduced antilipolytic effect, resulting in increased NEFA concentration, and then BHB, which in turn can have a direct negative effect on the endocrine pancreas. The lower the RQUICKI-BHB index and the higher the NEFA/insulin ratio, the more pronounced the insulin resistance. Insulin resistance indices are significantly determined by the values of NEFA and BHB [67]. Considering the direction of the regression line, we conclude that eHsp70 and TNF-α lead to the development of insulin resistance. In humans and laboratory mice with insulin resistance and T2DM, there is a lower concentration of iHsp70 and a higher concentration of eHsp70, both changes lead to an increase in TNF-α, and the higher the ratio in favor of eHsp70 the more pronounced the insulin resistance [20]. TNF-α shows a significant effect on the variability of the insulin resistance index even in healthy euglycemic subjects [68]. These results support our finding that both mediators are of great importance in the development of insulin resistance, with eHsp70 showing a significant correlation with the RQUICKI-BHB index only when controlled by TNF-α, while it correlates independently with NEFA/insulin ratio. TNF-α was found to be of great importance in the development of ketogenesis and the prediction of type II ketosis, which occurs in the same period of lactation as were cows in this experiment [45].

In our experiment, high eHsp70 levels in cows with high TNF-α levels were shown to have an additional negative effect on the adaptation of lipid metabolism and derived parameters. The intertwining of pathways, regulatory mechanisms, and effects of TNF-α and eHsp70 in the development of liver lipidosis, lipolysis, insulin resistance, and adipose tissue inflammation may be the reason for the additional adverse effects of eHsp70, which confirms the pro-inflammatory action of this chaperone in the extracellular environment. In the group of cows where there was an additional effect of eHsp70 (TNF-α + eHsp70 group) the concentration of BHB and NEFA was high enough to adversely affect all other aspects of metabolic adaptation of cows [69], which indirectly indicates the potential importance of eHsp70 in the overall metabolic adaptation of cows in early lactation.

Although the link between eHsp70 levels in the serum and lipid metabolite and lipid-based indexes and pro-inflammatory cytokine is strong, it remains unclear whether eHsp70 levels are a secondary feature of metabolic stress and inflammation in negative energy balance, or if eHsp70 plays a causal role in directly modulating lipid metabolism, metabolic stress an inflammatory response in cows during early lactation. 

## 4. Materials and Methods

### 4.1. Animals

The experiment included 50 s and third parity Holstein-Friesian cows kept in a free-stall system on a commercial dairy farm and an average milk yield of 6800–7900 kg per cow in standard lactation. The cows enrolled in the study were in the first week of lactation, clinically healthy, and had body condition scores (BCS) from 2.5 to 3.5 point on a scale of 1–5. On the farm, cows were fed twice daily using total mixed ration (TMR) and water was available *ad libitum*. Nutrient content of ration for experimental dairy cows in early lactation include: dry matter (DM) 21.5 kg; net energy of lactation 153.2 MJ; crude protein (CP) 18.3% DM; rumen undegradable protein 39.69% CP; fat 4.92% DM; fiber 17.2% DM; acid detergent fiber (ADF) 22.6% DM; neutral detergent fiber (NDF) 37.16% DM. Energy balance was evaluated according to body weight, offered meals, and average milk production using NRC standards [70]. The mean energy deficit was −26.5 ± 20.2 MJ/day). 

### 4.2. Blood Sampling and Laboratory Analysis

Blood samples were taken at the end of the first week after calving. Blood samples were collected 4 to 6 h after milking and feeding, from the coccigeal vein into evacuated tubes for serum separation. After clotting for 3 h at 4 °C and centrifugation (1500 G, 10 min), sera were immediately analyzed. All analyses were performed at the Laboratory of pathophysiology, Department of Veterinary Medicine University of Novi Sad. The following biochemical blood components were measured: concentration of serum eHsp70 was determined by ELISA colorimetric kit (Cusabio, Wuhan, China; Intra-Assay: CV < 15%; Inter-Assay: CV < 15%; assay sensitivity 1.25 ng/mL). using ELISA reader and washer by Rayto (Shenzhen, China); concentration of NEFA, BHB, cholesterol, triglycerides, HDL cholesterol, total bilirubin, albumin, and glucose was determined by colorimetric kit (Biosystem, Spain and Randox, Carlisle, UK) and spectrophotometer Chemray (Rayto, Shenzhen, China). VLDL concentration was calculated by the formula: VLDL = triglycerides/5. LDL cholesterol was calculated by the formula: LDL-C = total cholesterol − (HDL-C + triglycerides/5) [71]. Insulin concentration was measured by TOSOH AIA-360 (Japan). TNF-α was measured by standard kit manufactured by Cloud-Clone Corp (Wuhan, China; Intra-Assay: CV < 10%; Inter-Assay: CV < 12%; assay sensitivity 3.1 pg/mL) and Fluoroscan Ascent FL reader (Thermo Scientific, Waltham, USA).

### 4.3. Lipid Based Indexes of Functionality

The lipid-based index of functionality was calculated by the formulas: (1) Liver function index (LFI)—this index was determined based on formula LFI = (ALB − 17.71) / 1.08 + (CHOL − 2.57) / 0.43 − (TBIL − 4.01) / 1.21 [72]; (2) the liver lipidosis index was expressed as the NEFA to cholesterol ratio = NEFA/Cholesterol [38]; (3) the revised quantitative insulin sensitivity check index was calculated according to the formula—RQUICKI-BHB = 1 / [log (glucose) + log (insulin) + log (NEFA) + log (BHB)] [73]; (4) the anti-lipolytic effect of insulin was measured as NEFA to insulin ratio = NEFA/Insulin [74]. 

### 4.4. Statistics

To determine whether the negative energy balance affects the concentration of selected parameters and lipid-based functional indexes cows were divided into quartiles. The difference between cows with the lowest negative energy balance (lower quartile) and less pronounced negative energy balance (upper quartile) was confirmed using a *t*-test. Statistical analysis included Pearson correlation and regression line plotting between eHsp70 and other parameters, and TNF-α with other parameters. To confirm the pro-inflammatory effect of eHsp70, a partial correlation of eHsp70 with parameters was measured after the exclusion of TNF-α as a control factor. The additive effect of eHsp70 on the pro-inflammatory function of TNF-α was determined by the difference in blood parameters between the cows with a high concentration of TNF-α, and a high concentration of both TNF-α and eHsp70 using a *t*-test. SPSS Ver. 20 (SPSS, Chicago, FL, USA) was used for statistical analysis. Statistical significance was set at *p* < 0.05.

## 5. Conclusions

Due to the positive correlation between eHsp70 and TNF-α and their similar relations with lipid metabolites, we conclude that eHsp70 has pro-inflammatory effects implying lipolysis, fatty liver, and fat tissue insulin resistance. The eHsp70 showed an additive effect on the pro-inflammatory function of TNF-α on blood lipid parameters and functionality indexes based on blood lipid in dairy cows during early lactation. eHsp70 had an independent positive correlation with NEFA, but the correlation with BHB is under statistical control of TNF-α, which requires further research. eHsp70 may be a potentially interesting marker in the assessment of many aspects of inflammation-induced lipid metabolism change in early lactating cows.

## Figures and Tables

**Figure 1 metabolites-12-00019-f001:**
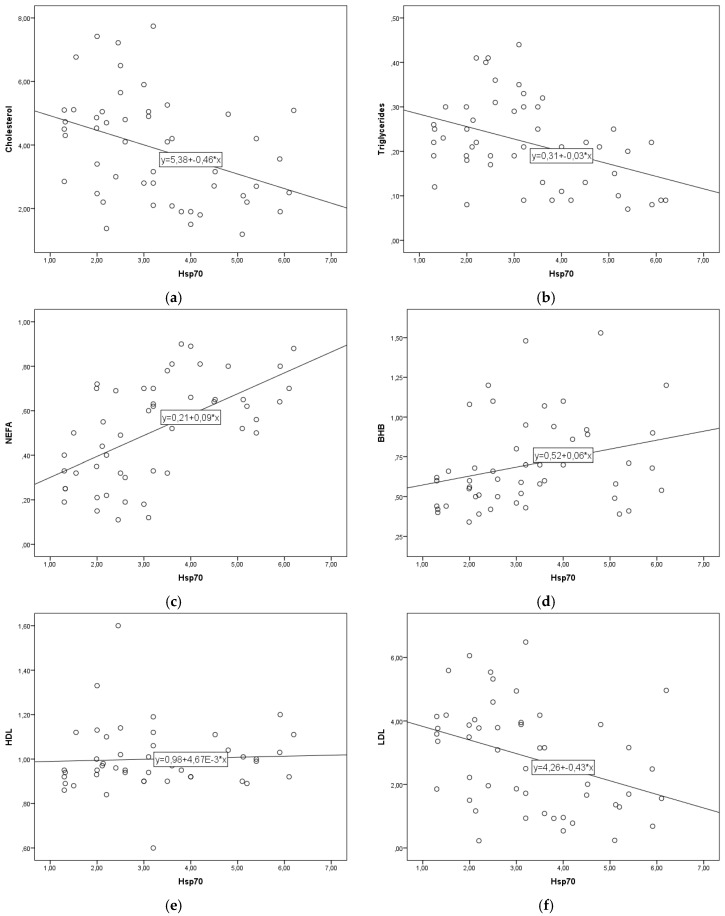
(**a**–**l**). Linear regression lines between eHsp70 and cholesterol (**a**), triglycerides (**b**), NEFA (**c**), BHB (**d**), HDL (**e**), LDL (**f**), VLDL (**g**), NEFA to cholesterol ratio (**h**), Liver functionality index (**i**), NEFA to insulin ratio (**j**), RQUICKIBHB (**k**), TNF-α (**l**) in cows during early lactation.

**Table 1 metabolites-12-00019-t001:** The value of metabolic parameters and its correlation with eHsp70.

Blood Parameters	Value of Whole Group	Value of Group with Energy Balance in Lower Quartile	Value of Group with Energy Balance in Upper Quartile	Coefficient of Correlation with eHsp70	*p* (Coefficient of Correlation)	*p* (Difference between Group of Energy Balance)
eHsp70 (ng/mL)	3.25 *±* 1.43	4.45 *±* 1.02	1.77 *±* 0.62	−0.093	NS	<0.05
Cholesterol (mmol/L)	3.89 *±* 1.68	2.32 *±* 0.42	5.09 *±* 1.31	−0.39	<0.01	<0.05
Triglycerides (mmol/L)	0.22 *±* 0.098	0.12 *±* 0.07	0.23 *±* 0.08	−0.408	<0.005	NS
NEFA (mmol/L)	0.51 *±* 0.23	0.71 *±* 0.11	0.35 *±* 0.12	0.587	<0.001	<0.05
BHB (mmol/L)	0.7 *±* 0.28	0.74 *±* 0.21	0.53 *±* 0.12	0.283	<0.05	<0.05
HDL (mmol/L)	0.98 *±* 0.19	1.01 *±* 0.13	1.09 *±* 0.09	0.047	NS	NS
LDL (mmol/L)	2.86 *±* 1.64	1.3 *±* 0.42	4.04 *±* 1.18	−0.372	<0.01	<0.05
VLDL (mmol/L)	0.04 *±* 0.019	0.02 *±* 0.01	0.05 *±* 0.011	−0.408	<0.005	<0.05
NEFA:Cholesterol ratio	0.18 *±* 0.14	0.31 *±* 0.09	0.08 *±* 0.05	0.516	<0.001	<0.05
Liver functionality index (LFI)	9.91 *±* 10.43	2.21 *±* 5.2	18.6 *±* 5.7	−0.403	<0.005	<0.05
RQUICKI-BHB	0.59 *±* 0.12	0.5 *±* 0.1	0.57 *±* 0.1	−0.312	<0.05	NS
NEFA:Insulin ratio	0.13 *±* 0.08	0.22 *±* 0.05	0.07 *±* 0.05	0.593	<0.001	<0.05
TNF-α (ng/mL)	10.07 *±* 3.1	12.47 *±* 1.92	7.29 *±* 1.29	0.443	<0.001	<0.05

NS—not significant.

**Table 2 metabolites-12-00019-t002:** Coefficient of correlation between lipid blood parameters and indexes with TNF-α, and partial correlation with eHsp70 after exclusion of TNF-α as control factor.

Blood Parameters	Coefficient of Correlation with TNF-α	*p*	Coefficient of Correlation with eHsp70 after Exclusion of TNF-α	*p*	Is Effect of eHsp70 Independent from Effect of TNF-α?
Cholesterol (mmol/L)	−0.433	<0.005	−0.243	NS	No
Triglycerides (mmol/L)	−0.373	<0.005	−0.292	<0.05	Yes
NEFA (mmol/L)	0.63	<0.001	0.447	<0.001	Yes
BHB (mmol/L)	0.317	<0.05	0.168	NS	No
HDL (mmol/L)	−0.211	NS	0.161	NS	N/A
LDL (mmol/L)	−0.403	<0.005	−0.236	NS	No
VLDL (mmol/L)	−0.373	<0.005	−0.292	<0.05	Yes
NEFA:Cholesterol ratio	0.519	<0.001	0.373	<0.005	Yes
LFI (Liver functionality index)	−0.384	<0.01	−0.289	<0.05	Yes
RQUICKI-BHB	−0.360	<0.05	−0.182	NS	No
NEFA:Insulin ratio	0.667	<0.001	0.445	<0.005	Yes

NS—not significant.

**Table 3 metabolites-12-00019-t003:** Difference in blood parameters in cows with high concentration of TNF-α and cows with high concentration of TNF-α + eHsp70.

Blood Parameters	High Concentration of TNF-α (n = 6)	High Concentration of TNF-α + eHsp70 (n = 7)	*p*
NEFA (mmol/L)	0.61 ± 0.09	0.82 ± 0.07	<0.05
BHB (mmol/L)	0.73 ± 0.08	0.95 ± 0.08	<0.05
Triglycerides (mmol/L)	0.19 ± 0.06	0.11 ± 0.05	<0.05
VLDL	0.042 ± 0.005	0.021 ± 0.005	<0.05
NEFA:Cholesterol	0.18 ± 0.05	0.35 ± 0.06	<0.05
NEFA:Insulin ratio	0.14 ± 0.07	0.22 ± 0.08	<0.05

## Data Availability

The data presented in this study are available in article.

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
