# Peer review of "The Correlation between Extracellular Heat Shock Protein 70 and Lipid Metabolism in a Ruminant Model"

_metabolites, 2021, doi:10.3390/metabo12010019_

Round 1

Reviewer 1 Report

Manuscript title: The correlation between extracellular heat shock protein 70 and

lipid metabolism in a ruminant model

In the manuscript, the authors investigated the relationships among extracellular heat shock protein 70 (eHsp70), inflammation and lipid metabolites in cows. In general, the authors have completed a reasonable study with very informative data on the relationships of metabolic parameters and eHsp70. Moreover, the statistical analysis and graphic presentation also have been completed in details. However, the Hsp70 has been well-known as one of the markers of inflammation. Most of the presentation has been only shown with the analysis results. The presentation of this study may be strengthened by taking a discussion and proposing the possible pathway to explain the cause and effect of eHsp70 formation.
The format of references should be revised in consistency, i.e. the capital letter and lower case in the reference title.
The reference should be added in the sentence “The half-life of chylomicrons and VLDL fractions of lipoproteins in the blood plasma of ruminants is very short, only 2 to 11 minutes, which is one of the reasons for their very low concentration in the blood. Therefore, HDL fractions accounted for 70 percent and LDL lipoprotein fractions for 20 percent of total blood lipoprotein”.

Author Response

In the manuscript, the authors investigated the relationships among extracellular heat shock protein 70 (eHsp70), inflammation and lipid metabolites in cows. In general, the authors have completed a reasonable study with very informative data on the relationships of metabolic parameters and eHsp70. Moreover, the statistical analysis and graphic presentation also have been completed in details. THANK YOU.

However, the Hsp70 has been well-known as one of the markers of inflammation. Most of the presentation has been only shown with the analysis results. The presentation of this study may be strengthened by taking a discussion and proposing the possible pathway to explain the cause and effect of eHsp70 formation.

THANK YOU. THIS IS THE FIRST REPORT IN DAIRY COWS WHERE, ON THE BASIS OF KNOWLEDGE FROM LITERATURE AND ANALOGY WITH THE INFLUENCE OF TNFα, WE SHOW THE PROINFLAMMATORY ACTION OF EXTRACELLULAR HSP70. DUE TO THE STATED FACT, THIS EXPERIMENTAL STUDY COULD BE CHARACTERIZED AS A PILOT RESEARCH. IN SUCH TYPES OF STUDIES, EXAMINING CORRELATIONS BETWEEN PARAMETERS IS THE FIRST STEP TOWARDS CONCLUSIONS. IN THE DISCUSSION, WE TRIED TO DESCRIBE POSSIBLE PATHWAYS THAT CONNECT HSP, FATTY TISSUE, HOMEORESIS IN EARLY LACTATION AND INFLAMMATORY FLOWS. WE ADDITIONALLY DISCUSSED THE RELATIONSHIP BETWEEN LIPOPROTEIN FRACTIONS AND CHAPERON, AND EXPRESION OF HSP INCLUDING NEW CITED REFERECES.

The format of references should be revised in consistency, i.e. the capital letter and lower case in the reference title.

THANK YOU.

The reference should be added in the sentence “The half-life of chylomicrons and VLDL fractions of lipoproteins in the blood plasma of ruminants is very short, only 2 to 11 minutes, which is one of the reasons for their very low concentration in the blood. Therefore, HDL fractions accounted for 70 percent and LDL lipoprotein fractions for 20 percent of total blood lipoprotein”.

THANK YOU.

Bergman EN, 1977, Disordes of carbonhydrate and fat metabolism. In Swenson, M. J. (ed). Duke s  physiology of dometic animals 9th. ed Cornel University Press, 357-367, London.

Palmquist, D. L. A kinetic concept of lipid transport in ruminants. A review. Journal of dairy science, 1976, 59.3: 355-363.

FINE/MINOR SPELL CHECKING WAS PERFORMED IN THE TEXT BY A PROFESSIONAL.

Reviewer 2 Report

Dear authors,

in my opinion you have done a very interesting scientific work. Stress proteins in the cell-environment relationship provide the cell with protection from harmful influences coming from the environment. Such research will surely be the focus of various medical and biochemical research for a long time to come. The obtained results can be used by the pharmaceutical industry to improve human health, but also by the economy to improve production and animal health. In order to improve the quality of your article, I would like to have a few questions to which I am asking for your answer:

  • The aim of the research is clear. However, the work hypothesis is not clearly stated. In my opinion, it is not enough to do correlation research alone. These correlations should be related to some parameter of animal health, its production or some other benefit. This will make the article much better and more interesting to the scientific community.
  • What were the environmental conditions in the barn where the cows were staying in the experiment (microclimatic parameters)? Whether the cows were on different farms or all 50 cows were on one farm? In what time frame did you take blood samples from cows?
  • Why did you do the research in the first week after calving? In my opinion, you should also have repeated measurements at a later stage of lactation. We would certainly get a better "picture" for the researched parameters.
  • Why did you choose Holstein cows? Maybe it would be much better if you chose a local, original breed?

The paper belongs to the scientific category, but needs to be supplemented with answers to the questions asked.

Author Response

Dear authors,

in my opinion you have done a very interesting scientific work. Stress proteins in the cell-environment relationship provide the cell with protection from harmful influences coming from the environment. Such research will surely be the focus of various medical and biochemical research for a long time to come. The obtained results can be used by the pharmaceutical industry to improve human health, but also by the economy to improve production and animal health. In order to improve the quality of your article, I would like to have a few questions to which I am asking for your answer:

THANK YOU.

  • The aim of the research is clear. However, the work hypothesis is not clearly stated. In my opinion, it is not enough to do correlation research alone. These correlations should be related to some parameter of animal health, its production or some other benefit. This will make the article much better and more interesting to the scientific community.

THANK YOU. ONE OF THE MOST IMPORTANT PARAMETERS IN EARLY LACTATION IN COWS IS ENERGY BALANCE. A LOWER ENERGY BALANCE IN THE FIRST WEEK OF LACTATION GIVES A HIGHER RISK OF DISEASE IN THE NEXT PERIOD. ENERGY BALANCE WAS CALCULATED FOR EACH OF THE COWS IN THE EXPERIMENT. ON THE BASIS OF THE VALUE OF ENERGY BALANCE, WE CLASSIFY COWS INTO TWO GROUPS - WITH BETTER AND BADER ENERGY STATUS AND THEN THE DIFFERENCE IN THE VALUE OF TESTED PARAMETERS BETWEEN THE TWO WAS DETERMINED. WHEN ONLINE SEARCHING FOR SCIENTIFIC RESULTS, THE USE OF NEGATIVE ENERGY BALANCE SENTENCE IS COMMON, WHICH MAY INCREASE THE VISIBILITY OF THE ARTICLE.

What were the environmental conditions in the barn where the cows were staying in the experiment (microclimatic parameters)? Whether the cows were on different farms or all 50 cows were on one farm? In what time frame did you take blood samples from cows?

CONSIDERING THAT THE COWS ARE VERY SENSITIVE TO HIGH ENVIRONMENTAL TEMPERATURES AND THAT THE HSP70 GROWS VERY FAST AFTER EXPOSURE TO THERMAL STRESS, THE EXAMINATION WAS CARRIED OUT IN THE THERMONEUTRAL PERIOD IN APRIL AND MARCH. WE MENTIONED THIS IN THE DISCUSSION AND CITED A SOURCE WHICH INDICATES THE HEAT LOAD OF COWS IN OUR REGION IN THAT PERIOD. BLOOD WAS TAKEN FROM ALL COWS DURING THE PERIOD. ALL COWS WERE FROM THE SAME FARM.

  • Why did you do the research in the first week after calving? In my opinion, you should also have repeated measurements at a later stage of lactation. We would certainly get a better "picture" for the researched parameters.

THE FIRST WEEK AFTER CALVING IS THE WEEK IN WHICH COWS ARE MOST LOADED WITH METABOLIC STRESS AND WHEN THERE IS A NEGATIVE ENERGY BALANCE, LIPOLYSIS AND GENERAL ADJUSTMENT OF METABOLISM TO LACTATION. THERE ARE THE MOST DRAMATIC METABOLIC DISORDERS THIS WEEK. DEVELOPMENT OF DISEASES AND REDUCED PRODUCTIVITY OF COWS DURING LACTATION ARE CONSEQUENCES OF INADEQUATE ADJUSTMENT OF METABOLISM IN THE FIRST AND POSSIBLY SECOND WEEK AFTER CALVING. IN THE LATER PERIODS OF LACTATION, THE ENERGY BALANCE IS STABILIZED AND THERE IS NO INFLAMMATION, LIPOLYSIS AND KETOGENESIS. HOWEVER, BY MAKING TWO SUBGROUPS BASED ON ENERGY BALANCE VALUES (RELATING TO YOUR FIRST COMMENT), WE CONCLUDE THAT THERE ARE DIFFERENCES IN METABOLIC ADAPTATION, WHICH (WE HOPE) IMPROVES "PICTURE" FOR THE RESEARCHED PARAMETERS.

  • Why did you choose Holstein cows? Maybe it would be much better if you chose a local, original breed?

THE HOLSTEIN-FRIESE COW BREED IS THE MOST COMMON BREED IN FARM CONDITIONS IN OUR REGION. INDIGENOUS COW BREEDS ARE NOT FARMLY ORGANIZED AND ARE BREAD EXTENSIVELY. THEY DON'T HAVE SUCH ECONOMIC SIGNIFICANCE. ON THE OTHER SIDE, THESE COWS DO NOT SHOW HOMEORESIS AND NEGATIVE ENERGY BALANCE DUE TO SMALL MILK PRODUCTION, SO THEREFORE THEY DO NOT SATISFY THE EXPERIMENTAL CONDITIONS. TESTING OF INDIGENOUS RACES IS OF GREAT BENEFIT WHEN TESTING THERMOTOLERANCE AND HSP70, BUT THIS WAS NOT THE TOPIC OF OUR INVESTIGATION. THIS RESEACH SHOWS THE CONNECTION OF EHSP70 WITH LIPIDS IN COWS DURING EARLY LACTATION.

The paper belongs to the scientific category, but needs to be supplemented with answers to the questions asked.

THANK YOU.

Reviewer 3 Report

The aim of the study was to determine the levels of extracellular heat shock protein 70 and its correlation with metabolic parameters in dairy cows. These are my comments to the authors:

  1. In my opinion introduction is well-written, nevertheless the authors should consider to make it a bit shorter.
  2. Material and methods seem to be dense. The authors should separate out some subsection and each method/methods should be descripted independently.
  3. How blood samples were collected? Were animals fasted before blood collection? What was the temperetaure/season during the experiments?
  4. Please provide full names of all kits that were used to determine all metabolic parameters.
  5. Were Elisa kits against eHsp70, TNFalpha and insulin specific against bovine proteins? Please provide this information in the manuscript. Furthermore, ELISA tests should be descripted in more detailed fashion. For example, intraCV and interCV values or assays sensitivity are missing.
  6. L183-185 – This sentence should be supported by reference.
  7. L192 “the enzyme sensitive lipoprotein lipase” Do the authors mean hormone sensitive lipase (HSL)? Please clarify this.
  8. In my opinion discussion is too long. There a lot of information regarding lipid metabolism in ruminants. However, the results should be discussed in the context of the aim of this work. Furthermore, the authors should list and discuses limitations of this manuscript.
  9. The authors should prepare the list of abbreviations which can be helpful for the readers.
  10. Can authors explain why they did not evaluate glucose levels?
  11. L225 There is a double dot at the end of the sentence.

Author Response

In my opinion introduction is well-written, nevertheless the authors should consider to make it a bit shorter. THANK YOU.

Material and methods seem to be dense. The authors should separate out some subsection and each method/methods should be descripted independently. THANK YOU.

How blood samples were collected? Were animals fasted before blood collection? What was the temperetaure/season during the experiments?

BLOOD SAMPLES WERE COLLECTED 4 TO 6 HOURS AFTER MILKING AND FEEDING, FROM THE COCCIGEAL VEIN INTO VACUUM TUBE FOR SERUM SEPARATION. FASTING IN DAIRY COWS IS NOT NECESSARY, IT SHOULD EVEN BE AVOIDED. IN HEALTHY COWS, A LARGE TIME BETWEEN MEALS CAUSES ACTIVATION OF LIPOLYSIS AND KETOGENEGESIS, SO THERE IS GREAT DIURNAL VARIATION OF METABOLITAL VALUES. TO AVOID THE PRANDIAL EFFECT AND TO REDUCE DIURNAL VARIATIONS, BLOOD WAS TAKEN 4 TO 6 HOURS AFTER THE MEAL. CONSIDERING THAT THE COWS ARE VERY SENSITIVE TO HIGH ENVIRONMENTAL TEMPERATURES AND THAT THE HSP70 GROWS VERY FAST AFTER EXPOSURE TO THERMAL STRESS, THE EXAMINATION WAS CARRIED OUT IN THE THERMONEUTRAL PERIOD IN APRIL AND MARCH. WE MENTIONED THIS IN THE DISCUSSION AND CITED A SOURCE WHICH INDICATES THE HEAT LOAD OF COWS IN OUR REGION IN THAT PERIOD. BLOOD WAS TAKEN FROM ALL COWS DURING THE PERIOD. ALL COWS WERE FROM THE SAME FARM.

Please provide full names of all kits that were used to determine all metabolic parameters. OK.

Were Elisa kits against eHsp70, TNFalpha and insulin specific against bovine proteins? Please provide this information in the manuscript. Furthermore, ELISA tests should be descripted in more detailed fashion. For example, intraCV and interCV values or assays sensitivity are missing. OK

L183-185 – This sentence should be supported by reference.

OK.

Bell AW, 1980, Lipid metabolism in the liver and selected tissues and in the whole body of ruminant animal, Progress in lipid Research, 18,177-179

Vernon RG, 1981, Lipid metabolism in the adipose tissue of ruminant animals, In: W. W. Christie (Ed.) Lipid Metabolism in Ruminant Animals. p 279, Pergamon Press, Oxford, U. K.

Shcwalm L, Shultz. H, 1976, Relationships of insulin concentrations to blood metabolites in the dairy cows, J Dairy Sci, 59, 2, 255-261.

L192 “the enzyme sensitive lipoprotein lipase” Do the authors mean hormone sensitive lipase (HSL)? Please clarify this. YES. WE MEAN HORMONE SENSITIVE LIPASE.

In my opinion discussion is too long. There a lot of information regarding lipid metabolism in ruminants. However, the results should be discussed in the context of the aim of this work. Furthermore, the authors should list and discuses limitations of this manuscript. THE DISCUSSION HAS BEEN UPDATED DUE TO OTHER REVIEWER COMENTS, BUT UNFORTUNATELY, WE CAN'T ASSESS WHICH PART WOULD BE SUITABLE FOR EJECTION. WE ARE READY TO SHORT THE DISCUSSION IF WE IF WE GET MORE SPECIFIC GUIDELINES, AND IF THIS IS EXTREMELY NECESSARY. LIMITATIONS ARE INCLUDED AT THE END OF DISCUSION.

The authors should prepare the list of abbreviations which can be helpful for the readers. OK.

Can authors explain why they did not evaluate glucose levels?

GLYCEMIA IS MEASURED FOR THE PURPOSE OF CALCULATING THE INSULIN RESISTANCE INDEX, BUT WE HAVE NOT PRESENTED IT AS A SEPARATE VARIABLE IN THE RESULTS. THE PURPOSE OF THIS PAPER IS TO EXAMINE THE RELATIONSHIP BETWEEN EHSP70 AND LIPID METABOLISM. LIPID METABOLISM AND GLUCOSE METABOLISM ARE NOT IN DIRECT RELATIONSHIP (AS IN NON-RUMINANT) BUT THEY ARE ALREADY CONNECTED INDIRECTLY THROUGH HOMERESIS ( MILK PRODUCTION, INFLAMMATION AND ENDOCRINE DISBALANCE). IN RUMINANRS, INSULIN RESISTANCE IS MUCH BETTER PARAMETERS OF GLUCOSE METABOLISM, THEN GLUCOSE CONCENTRATION ALONE. THAT IS WHY WE DIDN'T CONSIDER GLYCEMIA SPECIFICALLY.

L225 There is a double dot at the end of the sentence. OK

Round 2

Reviewer 2 Report

Dear authors,
thank you for the additional answers. I think we have now significantly improved the quality of the article. The answers are clear and accurate. You added new sentences in the article, they expanded the data in the tables, you added new literature.
Based on all of the above, I suggest an article for publication.

Reviewer 3 Report

I have no more comments for Authors.